# Effect of Impulsivity Traits on Food Choice within a Nudging Intervention

**DOI:** 10.3390/nu12051402

**Published:** 2020-05-14

**Authors:** Irene C. F. Marques, Megan Ting, Daniela Cedillo-Martínez, Federico J.A. Pérez-Cueto

**Affiliations:** Future Consumer Lab, Department of Food Science, Faculty of Science, University of Copenhagen, Rolighedsvej 26, 1958 Frederiksberg C, Denmark; sxp483@alumni.ku.dk (I.C.F.M.); kjg186@alumni.ku.dk (M.T.); fdk849@alumni.ku.dk (D.C.-M.)

**Keywords:** impulsivity, nudging, perceived variety, vegetable, visual presentation

## Abstract

Food choices are often driven by impulsive tendencies rather than rational consideration. Some individuals may find it more difficult resisting impulses related to unhealthy food choices, and low self-control and high impulsivity have been suggested to be linked to these behaviors. Recent shifts have been made towards developing strategies that target automatic processes of decision-making and focus on adjusting the environment, referred to as nudging interventions. Therefore, the purpose of this study was to investigate the effect of impulsivity traits on food choices within a nudging intervention (increased perceived variety). A total of 83 adults participated in an experimental study consisting of a self-service intelligent buffet. Impulsivity traits were measured using the UPPS-P impulsivity scale. General linear models were fitted to evaluate the effect of the five impulsivity traits on the difference of salad consumption (g) between the control and intervention situations. Results showed that impulsivity does not affect food choices in this nudging situation, suggesting that nudging works independently of the participant’s impulsivity score. Results also showed a significantly higher consumption of salad in the nudging versus the control setting (17.6 g, *p* < 0.05), suggesting that nudging interventions can be effective in significantly increasing total vegetable consumption across the whole impulsivity scale.

## 1. Introduction

The prevalence of obesity has continued to rise at an alarming rate and has now become one of the greatest public health challenges of the 21st century [1]. It is becoming more and more important to understand the factors that predispose individuals to making poor dietary choices that further contribute to overeating and obesity [2,3].

Foodscapes, defined as any environment where food is acquired, consumed, produced and prepared [4,5], play an important role in contributing to the growing obesogenic environment by influencing individuals to make unhealthier food choices [2,6]. The increase in availability and accessibility of ready-to-eat food options have resulted in a rising trend of individuals and families eating out more and cooking less homemade meals. Additionally, portion sizes have continued to grow incrementally over the years, all contributing to a higher consumption of more processed, energy dense, and less nutritious foods [7,8], excess energy consumption and lower intakes of fruits and vegetables [9,10,11].

Determinants of food choices are complex and involve a wide variety of factors [12,13,14]. The decisions and actions that lead to dietary habits and food choices are often based on internalized routines, often referred to as heuristics processes, that require minimal amount of active decision-making from the individual [6,12,13]. Because some individuals may find it more difficult in resisting impulses to consume more palatable, energy dense foods than others, it has been suggested that the traits of self-control and impulsivity lie at the root of these varying degrees of ability to resist these impulses and stop oneself from giving in to unhealthy food choices [15,16,17,18].

Self-control can be defined as “the ability to override or change one’s inner responses, as well as to interrupt undesired behavioral tendencies, such as impulses, and refrain from acting on them” [19]. Low self-control has often been equated to impulsiveness by researchers, although in principle, they are two different traits, which contribute independently to enacting behaviors [19,20]. Self-control has been shown to play a role in influencing food choices, as seen in various studies. These studies suggest that when self-control is low, individuals have the tendency to rely more on impulsive decision-making strategies such as external, environmental cues or heuristics [15,21,22,23]. Therefore, it can be suggested that food choices are mainly made through automatic processes, meaning they are driven more by impulsive tendencies rather than rational consideration.

Impulsivity is referred to as “the tendency to be spontaneous and act on intuition or heuristics” [24] or more specifically, “a predisposition toward rapid, unplanned reactions to internal or external stimuli without regard to negative consequences of these reactions” [25,26]. High impulsivity has been linked to several food choice and consumption behaviors such as unsuccessful dieting, frequent food cravings, binge eating, eating driven by external food cues, food addiction, and increased consumption of fast food or ready-to-eat food [17,25,27,28,29]. Increased impulsivity is thought to partly stem from impaired inhibitory control, which is “the ability to stop or suppress responses that are no longer required, inappropriate, or in conflict with current goals” [16,30]. It is therefore thought that individuals who are more impulsive have less ability to resist temptation of unhealthy foods that are more palatable due to preference for short-term immediate rewards and a higher sensitivity to external food cues [26]. It has been suggested that the combination of these various aspects of impulsivity and decreased inhibitory control might be associated with unhealthy food choices and thus contribute to overeating and ultimately to the overweight/obesity epidemic [17].

Because impulsivity is such a multifaceted behavior, a variety of tools are used to measure the three broad domains, which consist of impulsive personality traits (“dispositional tendencies toward impulsive behavior”), impulsive action (“deficits in behavioral inhibition”), and impulsive decision-making [17]. Impulsive personality traits are typically measured with self-reported questionnaires such as the UPPS-P Impulsive Behavior Scale or the Barratt Impulsiveness Scale (BIS), while impulsive action is typically measured with tasks such as the go/no-go task and impulsive decision making with the delay discounting task [16,17,31].

Originally developed in 2001 by Whiteside and Lynam and later adapted by Cyders and colleagues in 2007, the UPPS scale proposes impulsivity as a multi-dimensional construct, consisting of five impulsive personality traits: (1) negative urgency—“the tendency to act rashly when having negative emotions”, (2) lack of premeditation—“the tendency to act without thinking”, (3) lack of perseverance—“the inability to keep attention and motivation to complete tasks”, (4) sensation seeking—“the tendency to seek out and enjoy novel or exciting activities”, and (5) positive urgency—“the tendency to act rashly when experiencing positive emotions” [32,33]. Current literature shows that negative urgency and lack of perseverance are associated with overweight, obesity, and various eating disorders [28,29,34]. The purpose of developing the scale was to provide a consensus on which traits are measured across different existing impulsivity measures [35].

Due to the complex nature of the obesity epidemic, different strategies have been implemented to promote healthier food choices in individuals. In the past, the public health sector in Europe has mainly focused on targeting unhealthy dietary behavioral change via informational campaigns, legislation, and education, assuming that increasing individual knowledge will result in healthier food choice; however, these strategies have been shown to be only modestly successful [6,36,37]. Strategies targeting individual behavior changes in lifestyle have proven to be ineffective unless the change becomes a habit, but this is difficult to achieve as it requires a large amount of support and reinforcement in order to sustain the changed behavior [6].

Because food choices are often made automatically and based on heuristic processes [12,13,38,39], recent shifts have been made towards developing strategies that focus on adjusting the foodscape environment, rather than informing people what is the “right” or “healthy” choice [40,41]. Choice architecture can be defined as any modification to an environment with the aim of changing behavior in predictable ways using interventions. These interventions, called nudging interventions, are defined as “any aspect of the choice architecture that alters people’s behavior in a predictable way without restricting any options or significantly changing economic incentives such as time or money” [42]. Nudging interventions primarily work through automatic processes, not requiring the person to be fully engaging in rational thinking to fall into the nudge [41,43].

Typical nudging examples include altering placement (i.e., rearranging a cafeteria by placing less healthy options further away), increasing availability (i.e., increasing healthier options in vending machines), and altering presentation (i.e., presenting fruit and vegetables in an appealing way) [39,44]. Increasing perceived variety, which can be classified as a presentation type of nudging intervention [44], has proved to be an effective way of manipulating food choices in a predictable way [45,46]. According to previous studies, changing the presentation of salad components at a buffet setting by serving them in separate bowls, as opposed to one bowl, creates an illusion of increased variety of vegetables and increases vegetable intake among individuals [45,46].

Nudging interventions may have the potential to promote healthier choices specifically among highly impulsive individuals, who may rely more on automatic processes to make decisions [15,21,22,23]. To our current knowledge, there have been no studies investigating the effect of UPPS-P impulsivity traits on food intake within a nudging setting in a food lab or real-life setting. Considering the findings on low self-control and on how impulsivity is related to self-control [19,20], the present study aims at further understanding the relationship between the UPPS-P impulsivity traits and food choices, specifically under a nudging intervention. Based on certain study results [21,22,23], we expect higher impulsive individuals to rely more on heuristics when making decisions; therefore, we hypothesize that the higher an individual lies on the scale of impulsivity, the more their food choice will be affected by our nudging intervention. As only a couple of studies have investigated the effect of this specific nudging intervention, a secondary purpose of this study was to evaluate the effect of the nudging intervention, increasing perceived variety of vegetables, on increasing vegetable intake in a student population.

## 2. Materials and Methods

### 2.1. Sample and Recruitment

The study was conducted in October 2019 at the Evaluation Lab in the Department of Food Science at the University of Copenhagen, Frederiksberg campus. It consisted of a free lunch at a self-service buffet. To recruit participants, an event was created on social media containing information about the free lunch and the study. Participants were informed that the study’s general aim was related to food choice, but they were not informed of the specific study purpose. An informational flyer was also posted on the university’s kitchen lab bulletin board in one of the hallways of the Frederiksberg campus building. Participants could sign up for the buffet by contacting one of the three researchers with their preferred dates and schedules according to the available dates as listed on the informational page. Inclusion criteria were as follows: men and women (age ≥ 18) with no food allergies. No incentives were used aside from providing food free of charge.

### 2.2. Questionnaire Development

Each participant was given a questionnaire consisting of 4 different sections. The first 3 sections were to be answered prior to the meal, while the last section was to be answered post meal consumption (see Appendix A for the complete questionnaire).

Section 1 contained sociodemographic questions regarding gender, age, occupation, highest level of education, and self-reported height (cm) and weight (kg), which were later used to calculate Body Mass Index (BMI) of each participant. Additional questions included food consumption patterns (omnivore, flexitarian, vegetarian, pescatarian or vegan) and any recent changes in lifestyle habits in the past 2 months (diet, exercise, smoking).

Section 2 contained the short version of the UPPS-P Impulsive Behavior Scale questionnaire consisting of 20 statements. The original UPPS-P contains 59 items and can be time-consuming for participants, therefore a short version was created in English in 2013 and validated as a reliable alternative in 2014, as the Crohnbach’s alpha values for all the impulsivity traits are >0.7 [47]. Each statement in the survey is rated from “agree strongly” (1) to “disagree strongly” (4). The higher the scores, the more impulsive the person tends to be [35].

In Section 3, two questions were asked using a 10-point Likert scale regarding participants’ level of hunger (1 = starving, 10 = extremely full) and how much they felt capable of eating (1 = nothing at all, 10 = a lot). Section 4 was administered after the meal and included two 10-point Likert scale questions measuring level of satiation (1 = starving, 10 = extremely full) and liking of the food (1 = not at all, 10 = a lot). Participants only answered Section 1 and Section 2 on the first visit.

### 2.3. Equipment

An intelligent buffet (ibuffet) was used to measure the amount of food that each participant served themselves at the University of Copenhagen’s Future Consumer Lab. This equipment simulated a typical buffet table with 4 serving units, each containing an integrated scale not visible to participants and a radio-frequency identification (RFID) reader, as described elsewhere [48]. Wristbands containing RFID were assigned to a code and then distributed to each participant. Participants were asked to check-in at each serving unit using their wristband every time they served themselves food. Data regarding the amount of food each participant served themselves was obtained through the coded RFID bracelet. Equipment was calibrated before each lunch session throughout the day.

### 2.4. Study Design

The study was designed as a one factor experiment (condition: control vs. nudge) in which the order was counterbalanced across participants. Participants were not informed which setting (control or nudge) each date was assigned. The study lasted a total of four days spread over a three-week period, with two days for the control setting and two days for the nudging setting. Each day consisted of four eating sessions, which lasted 45 min each. The ibuffet was placed against the wall and cutlery and plates (27 cm in diameter) were placed on a table to the left of it. Olive oil, balsamic vinegar, salt, pepper and chili flakes were also placed at that table, where participants could add them to their food freely. Glass water jugs (1.2 L) and glasses (250 mL) were placed at each table. Four tables, each with four chairs, were set up in the room, allowing up to 16 participants per session. Participants were informed that they could serve themselves from the buffet ad libitum and return as many times as they liked. Food on the buffet was refilled during each session as needed to ensure that plenty of food was available.

Food offered during the control setting consisted of a mixed salad composed of 7 vegetables (roasted broccoli, roasted cauliflower, cucumber, lettuce, tomato, dressed white cabbage and roasted zucchini), placed at the first serving unit in a 6 L green bowl. Pasta was served on the second unit in a 6 L red bowl, followed by vegetable sauce (composed of 50% mushrooms, eggplant, carrots, broccoli and 50% crushed-can tomatoes and spices: powdered onion, basil leaves, rosemary, dried oregano, chili flakes, vegetable stock cubes, salt, sugar and pepper) and lastly, the meat sauce (composed of 30% ground beef, 20% vegetables—same ones as in the vegetable sauce, and 50% tomato sauce as mentioned above). Both sauces were identical in aspect and served in 2.8 L rectangular porcelain containers and placed on warm serving units. Salad was served with tongs and the rest of the meal components with large serving spoons (50 mL). A diagram of the control setting can be seen in Figure 1.

During the nudging setting, the same food and its display as in the control was offered except for the presentation of the mixed salad. Instead, mixed salad components were separated and served in individual transparent bowls, allowing participants to create their own salad, although the variety of vegetables was the exact same as in the control. Cabbage and lettuce were served together in a relatively larger bowl (1.5 L) and the remaining vegetables were served in slightly smaller bowls (1 L): tomato, cucumber, and cauliflower alone, and broccoli and zucchini in the same container. Regular tablespoons (approx. 15 mL) were used to serve each of the vegetables and a small tong was used for the white cabbage and lettuce. The bowls were placed on the same serving unit, in order that the quantity served from each vegetable bowl would be totaled into one final measurement, “salad”. As previously mentioned, this type of nudging is known as increased perceived variety. Figure 2 depicts the diagram presentation of the nudging intervention setting.

### 2.5. Data Analysis

A total of 99 participants signed up for the two settings of the buffet. Complete-case analysis was carried out with a total of 83 participants. Due to one participant leaving the “weight” question blank, one weight value was completed using the median value of the total sample.

Statistical analysis was performed on R Studio software (version 1.2.1335). Data distribution was determined with histograms, residual and QQ plots. Descriptive statistics, such as total counts (N), percentage of total sample, and mean and standard deviation (SD) were used to characterize participants. Consumption of the meal components is reported using mean and SD for both the control and nudging settings. Mixed model ANOVA was performed to find significant differences in consumption between the two settings for each meal component, controlling for the effect of confounding factors such as gender, BMI, other meal components, and differences in lunch settings among participants (1st session setting—nudge vs. control; schedule and interval between sessions). General linear regression models were used to test the association between the 5 impulsivity traits and the difference in salad consumption between the two settings. Significant values were considered when the p-value was <0.05.

The energy content (kcal) of each of the meal components was calculated using Matportalen [49]. It was used to compare total calories consumed during the control setting and the nudging setting, this information can be found in Appendix A.

### 2.6. Ethical Approval and Participants’ Consent

Participants voluntarily signed a consent form before taking part in the buffet, where it was stated that data collected would be kept confidential and anonymous and would be used solely for the purpose of the study. Participants were able to opt out of the study at any time by notifying any of the three researchers. Data was stored according to General Data Protection Regulation. The Research Ethics Committee for Health and Science at the University of Copenhagen approved the study protocol (Ref. 504-0107/19-5000).

## 3. Results

Although 99 participants signed up for the study, 16 were excluded from data analysis due to either missing one buffet setting (N = 5) or missing both settings (N = 11). A final sample with a total of 83 participants was obtained (16% dropout rate). Sociodemographic distribution and the impulsivity scores of the final sample (N = 83) are shown in Table 1. The sample consisted mainly of females (70%). Participants were mostly students (84%), with the majority completing a bachelor’s degree (63%). Concerning diet, more than half of the sample were omnivores (55%), followed by flexitarians (24%) and vegetarians (13%). Impulsivity scores of the sample population showed higher scores in the “sensation seeking” trait (mean = 2.8) and lower impulsive scores in the “lack of premeditation” trait (mean = 1.8). The scores obtained for the self-reported hunger were similar at both the first and second visit of each participant (mean of 3.5 and 3.8, respectively), therefore we can conclude that there was no difference in hunger for both the nudge and control setting.

Table 2 displays the mean and SD quantity, in grams (g), of all self-served meal components consumed at the buffet. According to the mixed methods ANOVA test, exposure to the nudge significantly increased salad consumption compared with the control (17,6g, *p* = 0.03). Not reported here, similar results were obtained when using a paired Student’s t-test (only salad consumption was significantly different between the settings *p* = 0.02).

In the mixed methods ANOVA, BMI affected only salad consumption (*p* < 0.01), while gender influenced the models for the remaining meal components consumption, total consumption, and total Kcal, but did not influence salad consumption. Additionally, all meal components significantly affected the meal component being tested.

Differences in salad consumption had both positive and negative values (see Appendix A), signifying that our nudging design was effective on some participants (those with positive values), but also had an opposite effect on other participants (those with negative values). We observed that none of our population sample scored high (score of 4) for the traits of Lack of Premeditation, Lack of Perseverance, and Positive Urgency. A few participants scored high in the Negative Urgency and Sensation Seeking trait. No participants scored low (score of 1) in the Sensation Seeking trait.

Linear regression models were performed to analyze the association between the impulsivity traits and the difference of salad consumption between the two lunch settings (nudge minus control). As seen in Table 3, we found no significant results for the various models applied, neither on the unadjusted nor on the adjusted models. Several models were developed, with different combinations of variables (not all were reported here since no additional knowledge was gained from it).

## 4. Discussion

The primary purpose of this study was to evaluate the effect of the UPPS-P impulsivity traits on food choices within a nudging setting. In order to evaluate this, we performed various general linear regressions models using participants’ impulsivity scores and change in salad intake (in grams). No significant association was found between the designed nudged and any of the impulsivity traits (Table 3); therefore, we reject our hypothesis and can conclude that nudging had the same effect regardless of the participant’s impulsivity score derived from the short UPPS-P Impulsivity Behavior scale.

To our knowledge, there have not been any studies investigating the effect of impulsivity traits on food choices within a nudging setting. However, considering how researchers often equate low self-control with high impulsivity [20], studies of similar design have measured the effects of self-control on food choices. Three studies found that self-control moderated the effect between a nudging intervention (which they refer to as an influence/social/scarcity heuristic) and food choice. One study found that people made fewer healthy choices under low self-control when no nudging intervention (social proof heuristic) was present, but that when the nudging intervention was present, people made healthier decisions under low self-control [21]. A similar study also found that participants with lower self-control were more likely to buy the healthier food option of low-fat cheese when it was associated with the social proof heuristic [22]. Another similar study found that the number of healthy food choices increased as self-control levels decreased, but only in the presence of a scarcity heuristic [23]. These findings all suggest that individuals with lower self-control may actually benefit more from certain nudging interventions that target heuristics compared to individuals with higher self-control.

Although our hypothesis was in line with the previously mentioned studies’ results, ours did not reveal the same. Our findings, however, were similar to one study which investigated whether a “proximity effect” nudging intervention (which increased the distance between the unhealthy snack food and the individual) was moderated by cognitive resource. Although this study was based on previous controversial associations made between lower cognitive resource and lower impulse control, their results showed [50,51] that participants were likely to take the snacks regardless of their level of cognitive resource, concluding that cognitive resource did not moderate the effect of the nudging intervention [52]. These results are similar to our findings where impulsivity did not moderate the effect of the nudging intervention. Interestingly, another study [15] found that self-control moderated the effect between calorie labeling and food choices in a young adult population, showing that the nudging intervention had a stronger effect in individuals with high self-control (equivalent to lower impulsivity).

A secondary aim of this study was to evaluate the effect of increasing perceived variety of vegetables, on increasing vegetable intake in a student population. Our study found that increasing perceived variety significantly increased the amount of salad consumption by 17.6 g (*p* = 0.03, but it did not significantly decrease any of the other meal component consumption (pasta, vegetable or meat sauces). No significant difference was found in total intake (g) or total energy intake (kcals) between the two settings (Table 2), which is partially in alignment with previous reports that used the same type of nudge. One similar study showed that increased perceived variety significantly decreased meat consumption (g), total consumption (g), and total energy intake (kcal), but no significant increase in vegetable consumption was found [45]. Another similar study conducted in males only, found that increasing perceived variety significantly increased salad consumption (g) and significantly decreased pasta consumption (g) [46]. The same study did not see any significant difference in total consumption (g) between the nudge and control setting although total energy intake (kcal) was significantly decreased in the nudging setting [10]. Our study only saw a significant difference in salad consumption and not in other meal components, which is similar to another study’s findings showing increased variety significantly increasing vegetable intake (g) but not total energy intake (kcal) [53]. As salad was the least energy dense dish among our served meal components (see Appendix A), a much larger difference in salad intake would have been needed in order to see a significant change in total energy intake as the previous study also suggests [53].

The current study has several limitations and strengths. One limitation was that our sample population was not generalizable as it consisted mainly of students and females. Additionally, the design of this study was based on the statistical power of previous studies on increased perceived variety [45,46]. Our study’s effect size was smaller due to the weaker difference in salad consumption, possibly resulting from the type of nudge that was chosen (Cohen’s *d* = 0.21, for salad consumption). Systematic reviews of various nudging interventions suggest that depending on the type of nudge implemented, different levels of effectiveness can be achieved [48,54,55,56]. Previous nudging interventions reporting on the effect of increased perceived variety used more than one serving station and, in one study, multiple nudging strategies. This might have resulted in synergetic effects and larger differences in consumption between control and nudge settings [46,55]. Our study focused only on one specific nudge design and used only one buffet station, which may be a reason our study did not see as substantial results.

Another possible reason as to why we did not see as high increases in salad consumption as the previous studies was the poor choice of serving utensils used for the different salad components during the nudge setting (regular tablespoons). Based on our observation, participants appeared to have had some trouble when serving the individual salad components with the tablespoons compared to using the tongs for the mixed salad in the control intervention, which potentially discouraged them from serving more vegetables. The authors recognize that this poor choice in utensils could have possibly influenced the results of salad consumption in the study; however, an increase in salad consumption in the nudging setting was seen regardless of this choice.

One potential limitation was that by having the study take place in an experimental food setting, participants might have been conscious about being evaluated, possibly affecting their intake and influencing the results [57]. Other external factors, such as price and variety, which are found in real-life food choice settings, can also alter people’s choice, but these factors were not simulated in our study [58].

Another potential limitation of this study was the fact that food choice does not necessarily equate to food consumption, as we used an intelligent buffet to measure the total amount of self-served meal components [47]. However, based on observation, no plate waste was left from participants in our study, indicating that the amount of self-served meal components was equivalent to actual consumption.

Future studies can benefit from improving on the limitations of this study by improving the study design. Including a wider variety of participants to make the population more generalizable, conducting the experiment in a real-life food choice setting such as a cafeteria, including more participants, and choosing a nudging intervention or combining nudging interventions that have been shown to have larger effects may all contribute to a stronger study. Additionally, using different measures of impulsivity may be interesting to explore, as impulsivity is such a complex construct.

Of note, this study also found that BMI affected salad consumption (*p* < 0.01), meaning that the higher the BMI, the more salad participants consumed in the nudging setting. Since there are some studies associating BMI with unhealthy food choice behaviors and high impulsivity in certain traits, [17,27,28,29] it could be interesting for future studies to look into this, as this is outside our study’s scope.

A strength of our study was that the ibuffet used was able to record the precise quantities of food served by each participant, eliminating the need to perform time-consuming manual weighing of food or to rely on self-reported estimates, the latter being an unreliable estimate of self-served food and intake [39]. Additionally, participants were not aware that their servings were being measured, as the ibuffet contained an enclosed hidden scale, which reduced bias. Additionally, our menu included a vegetarian option, resembling the campus canteen choices and, thus, a “real life situation” more closely.

## 5. Conclusions

To the best of our knowledge, our study was the first experiment analyzing the effect of impulsivity measures on food choices performed with or without a nudge. Our preliminary findings provide empirical evidence supporting that food choices within a nudging setting are independent of impulsivity, showing that increasing perceived variety can be effective in significantly increasing total vegetable consumption in university students and across the whole impulsivity scale. Our results also suggest that it may not be necessary to design specific nudging interventions tailoring to people with higher levels of certain impulsive traits (negative urgency and lack of perseverance), but further studies are needed to validate our results.

## Figures and Tables

**Figure 1 nutrients-12-01402-f001:**
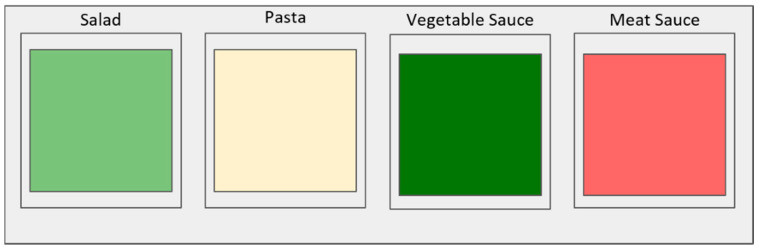
Diagram of the control setting.

**Figure 2 nutrients-12-01402-f002:**
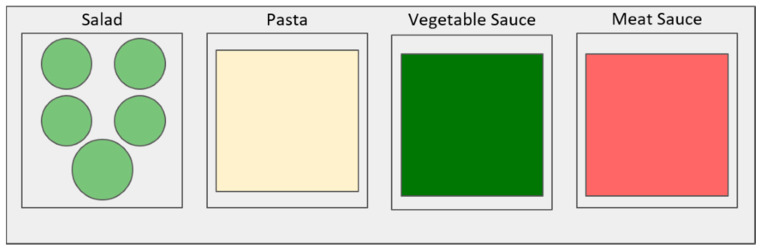
Diagram of the nudging setting.

**Table 1 nutrients-12-01402-t001:** Characterization of total sample (N = 83).

	N (%)		Mean (SD)
***Gender***		***Age***	24.6 (3.5)
**Female**	58 (69.9)		
**Male**	25 (30.1)	***Hunger scores***	
***Education***		First visit	3.5 (1.1)
**High school or lower**	10 (12.0)	Second visit	3.8 (1.2)
**Bachelor**	52 (62.7)	***Impulsivity scores***	
**Master**	21 (25.3)	Lack of Premeditation	1.8 (0.5)
***Occupation***		Lack of Perseverance	1.9 (0.5)
**Student**	70 (84.3)	Positive Urgency	1.9 (0.6)
**Employed**	3 (3.6)	Negative Urgency	2.1 (0.7)
**Unemployed**	10 (12)	Sensation Seeking	2.8 (0.6)
***Diet***			
**Omnivore**	46 (55.4)		
**Flexitarian**	20 (24.1)		
**Pescatarian**	4 (4.8)		
**Vegetarian**	11 (13.3)		
**I do not know**	2 (2.4)		
***BMI***			
**Underweight (<18.5)**	6 (7.2)		
**Normal weight (18.5–24.9)**	63 (75.9)		
**Overweight (25–29.9)**	12 (14.5)		
**Obese (>30)**	2 (2.4)		

N—total counts; %—percentage of total sample; SD—standard deviation.

**Table 2 nutrients-12-01402-t002:** Mean and SD quantity of each self-served meal components, total consumption, and total calories.

Variables	Control Mean (SD)	Nudge Mean (SD)	*p* ^1^
**Salad (g)**	151.7 (87.0)	169.3 (82.9)	**0.03**
**Pasta (g)**	208.8 (128.1)	210.8 (118.6)	0.35
**Vegetable sauce (g)**	135.0 (116.6)	137.2 (119.0)	0.40
**Meat sauce (g)**	166.1 (160.0)	164.5 (155.8)	0.41
**Veg + meat sauce (g)**	301.1 (143.0)	301.7 (141.3)	0.35
**Total consumption (g)**	661.6 (242.1)	681.8 (235.6)	0.32
**Total calories (kcal)**	435.2 (211.6)	438.9 (198.8)	0.81

^1^ Mixed models ANOVA *p*-values, controlled for gender, BMI and other meal components, and randomized for participant, 1st session setting—nudge or control, lunch schedule, and days of interval between sessions; a p-value < 0.05 indicates significant differences between settings.

**Table 3 nutrients-12-01402-t003:** Linear regression models for the impulsivity trait and the difference of salad consumption (g) eaten between the nudging and the control setting.

	Lack of Premeditation		Lack of Perseverance		Positive Urgency		Negative Urgency		Sensation Seeking	
	**slope ± SE**	***p***	**slope ± SE**	***p***	**slope ± SE**	***p***	**slope ± SE**	***p***	**slope ± SE**	***p***
**1**	3 ± 16	*0.84*	-11 ± 15	*0.48*	14 ± 15	*0.34*	17 ± 12	*0.18*	-3 ± 13	*0.77*
**2**	3 ± 16	*0.85*	-12 ± 16	*0.45*	12 ± 15	*0.40*	19 ± 13	*0.14*	-4 ± 14	*0.74*
**3**	0 ± 16	*1.00*	-11 ± 15	*0.49*	9 ± 14	*0.55*	17 ± 13	*0.20*	-1 ± 13	*0.93*
**4**	1 ± 18	*0.96*	-20 ± 17	*0.26*	13 ± 16	*0.41*	18 ± 14	*0.21*	7 ± 14	*0.65*
**5**	-5 ± 17	*0.77*	-20 ± 16	*0.21*	13 ± 15	*0.36*	20 ± 13	*0.13*	-1 ± 14	*0.93*
**6**	-9 ± 18	*0.62*	-21 ± 16	*0.20*	10 ± 16	*0.50*	19 ± 13	*0.16*	3 ± 15	*0.84*

SE: Standard Error; p: p-values of the linear regression model; 1: unadjusted; 2: adjusted for age and gender; 3: adjusted for age, gender, and BMI; 4: adjusted for age, gender, BMI, education, occupation, type of diet, and recent changes in lifestyle habits; 5: adjusted for 1st session setting—nudge vs. control, lunch schedule, and days of interval between sessions; 6: adjusted for 1st session setting—nudge vs. control -, lunch schedule, days of interval between sessions, age, gender, and BMI; Note: additional models were developed, not reported here since no further knowledge was gained from it.

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
