# Peer review of "Effect of Impulsivity Traits on Food Choice within a Nudging Intervention"

_nutrients, 2020, doi:10.3390/nu12051402_

Round 1

Reviewer 1 Report

The authors report on a 'perceived variety' nudge in a self-service buffet, and investigate whether impulsivity moderates effectiveness. 

The manuscript is well-written and easy to follow, and the study is timely as it  investigates one of the core hypothesized working mechanisms of nudges (as making use of automatic tendencies) in a realistic environment with real behavior. I would like to compliment the authors on this work. I do, however, have a number of issues that I like to address:

  1. The formulation of the hypotheses can be made more precise. In the final paragraph of the introduction it is first stated that "nudging may also be an effective intervention in decreasing overall energy consumption in highly impulsive individuals". The hypothesis as formulated in the next sentence specifies that the nudge should actually be more effective in this population (rather than also effective). I would like to see this clarified. On top of that, I would like to get more insight into what the authors exactly expect when they talk about a greater effect. Do the authors expect that highly impulsive individuals normally make less healthy choices and that their nudge buffers against this such that highly impulsive individuals make equally healthy decisions in the nudge condition, or do they expect that they actually make more healthy decisions in the nudge condition. This needs to be clarified. Besides, the theoretical basis for the hypothesis can be improved. The hypothesis should be supported more with literature. The paragraph where the hypotheses are spelled out does not include any references, and some new claims are made that were not make before in the introduction. Theses need to be supported with literature, or at least be explained better. 
  2. I would like to see a more precise (working) definition of nudging. The authors refer to the classic definition of Thaler and Sunstein, but don't elaborate on it. An example of a typical nudge would help here. Moreover, I would like to see more about the specific nudge used in this study (perceived variety). How does this fit the definition of nudging? What is already known about this nudge? And how is it supposed to work? It is unfortunate that the authors did not measure perceived variety but only infer this from the manipulation, but perhaps the authors could speculate about this further in the discussion section. As the authors point out in the discussion section, the nudge in a way also increases effort to obtain vegetables, which is counterintuitive with the idea of nudges making it easy to perform the desirable (in this case healthy) behavior. 
  3. How does impulsivity relate to other correlates of automatic decision making, such as lower cognitive resources or low self-control. Although the hypothesis that nudges should be more effective for highly impulsive people follows from the assumptions of nudges, empirical evidence from other correlates of automatic decision making does not seem to support this hypothesis (e.g., Hunter, Hollands, Couturier, & Marteau, 2018)
  4. Results: The authors do not report any test statistics or effect sizes. I would like to see those reported. Besides, the authors control for a lot of possibly confounding variables. However, it is unclear whether this is really necessary. In the discussion section the authors claim to have high power, but by putting this many variables in the model, power decreases. Do those control variables correlate with the main DV? And do results (reported in Table 2) change if less control variables are included?
  5. Does impulsivity relate to consumption in absolute terms? The authors currently only report on the difference in consumption between the two conditions, but it would also be interesting to report if impulsivity is related to consumption to begin with. 
  6. I would like the authors to elaborate a bit further on what their results imply and how they relate to previous literature in the discussion section. For example, what does it mean that there's no difference in total energy intake? How do these results relate to other studies on the perceived variety nudge? And how do these results relate to other studies that have used other nudges and/or other indices of automatic behavior?

Minor comments:

  • line 93: the word 'unconscious' is redundant and not necessarily supported by research. People may not always notice the nudge, but that does not necessarily imply that they unconsciously choose healthier food. 
  • The design of the study could be formulated more precisely. If correctly understood, it is a 1 factor (condition: control vs. nudge) within subjects design with the order of the conditions counterbalanced between participants. 
  • Some measures can be more accurately described. For example, could the authors provide more detail about the formulation of the question on recent changes in lifestyle habits? And could the authors give an example item of the UPPS-P and report on reliability?
  • Line 163: The word 'intervention' should be replaced with 'study'. Intervention can be confused with nudge intervention.
  • Line 233: Could the authors provide the test statistics for this?
  • The scatter plots of Figure 3 are not necessary informative and could be removed from the manuscript. 
  • In the discussion section it is mentioned that the time interval that varied between participants could be a limitation of the study. I think it should be possible to test whether this actually affected the main DV. What do the data say about this possible limitation? 

Reviewer 2 Report

Review of Nutrients 744630

The authors introduce an interesting study on the interplay between impulsivity and the effectiveness of nudging interventions on food intake. This is a timely topic considering the rise of nudging studies in the last decade, and the need for more insight into boundary or moderator conditions on nudge effectiveness.

I have a few comments and suggestions on the current version of the manuscript that I will list below.

Throughout the introduction, there is need for more definitions and a bit more elaboration on certain statements. For example, what is a foodscape? How is the food environment defined? What does development in technology have to do with higher consumption of certain foods?

The authors mention that our food choices are often based on routines, and then seamlessly continue with terms like ‘mindlessly’, and ‘impulsive tendencies’. However, these three things (routines, mindlessness, and impulsivity), are not the same, and should be treated and defined more distinctly. I would also like to see more references for this particular statement on mindless food choices, as the Wansink paper might not be strong enough as a main pillar of support, considering recent criticism on the methodological and statistical rigor involved in this particular lab’s published work, and the Kroese et al. paper is actually a field experiment that did not show that people make impulsive choices more than rational ones, per se.

There is probably a reason why the authors choose to not introduce nudging as nudging but as ‘food interventions’, but this is confusing, as informational campaigns can also be considered food interventions. As the authors continue with a nudging study, using a nudging intervention, I would advise to leave this extra step out of the manuscript and introduce nudging earlier on.

Choice architecture is not interchangeable with the term nudging. The term nudging means influencing the choice architecture in a certain way.

The hypothesis on impulsivity is very interesting – I do wonder how the authors few ‘internal’ cues for impulsivity in this regard. What if high impulsive people are driven by pressing internal cues or drives feeding into their behavior, would nudging then still be more effective then with low impulsivity?

As the authors use an atypical nudge in their study, there needs to be more information on how this feeds into heuristic or impulsive decision making.

Finally, and perhaps most importantly, as the authors observed trouble with cutlery and serving spoons, this might have severely interfered with their intended study aims. As such, the results are very difficult to meaningfully interpret. The within-participants design, while having advantages, may also very well have resulted in participants noticing the nudge and being cautious when serving themselves, knowing they were in a food choice study. Taken together, this is a lot of noise that may have overwritten any differences between conditions. As the authors also did not include any questions on how the participants perceived and experienced the different settings, it is impossible to make valid conclusions about nudging, or impulsivity, based on this study.

Round 2

Reviewer 2 Report

The manuscript has benefited greatly from this review round - the Introduction is much clearer and contains more of the relevant literature, as does the Discussion.

However, the use of different serving utensils in the experimental and control condition, and the reported difficulties with those utensils, in combination with a single-study report call for more research and stricter tests of the author's hypothesis before conclusions about impulsivity and nudge effects can be drawn. The argument that a nudge effect was observed regardless of the serving spoon issue does not hold - it is impossible to tell whether this effect is regardless or because of the noise caused by the design.